# Developing a priority global research agenda for antimicrobial resistance in the human health sector: protocol for a scoping review

Raph L Hamers [1,2] Alessandro Cassini,[3] Koe Stella Asadinia,[1]
Silvia Bertagnolio[3]

¹Eijkman-Oxford Clinical Research Unit, Jakarta, Indonesia
²Centre for Tropical Medicine and Global Health, Nuffield Department of Medicine, University of Oxford, Oxford, UK
³Department of Surveillance, Prevention and Control, Division of Antimicrobial Resistance, World Health Organization, Geneva, Switzerland

**Correspondence to**
Dr Silvia Bertagnolio;
bertagnolios@who.int

## ABSTRACT

**Introduction** To accelerate the response to the public health threat by antimicrobial resistance (AMR), the WHO is developing a Global Research Agenda for AMR in the human health sector that aims to provide a global and transparent assessment of priority knowledge gaps related to critical bacteria—including *Mycobacterium tuberculosis*—and fungi that inform control and response strategies to tackle AMR by 2030. A literature scoping review represents the first phase in a stepwise process, and we hereby outline the protocol to review current knowledge gaps and research questions on AMR in the human health sector.

**Methods and analysis** This literature scoping review will follow the Arksey and O'Malley (2005) methodology and will include: (1) a hand search to identify relevant WHO guidelines and documents suggested by the WHO Steering Group for the AMR Global Research Agenda; (2) a grey literature search through a stakeholder mapping process and google searches of organisational websites; (3) a systematic search of relevant systematic reviews through bibliographic databases (PubMed, Embase and Web of Science); (4) screening of the reference lists of included studies. We will include relevant publications from the last 10 years (January 2012 to December 2021). Two researchers separately will review the yielded citations to determine eligibility based on predefined criteria. Relevant research questions with attributes will be extracted using a tool developed through an iterative process by the research team. Each identified research question will be classified and aggregated according to a conceptual framework (ie, 'knowledge matrix'), composed of three themes (ie, Prevention, Diagnosis and Care & Treatment) and four cross-cutting domains (ie, Descriptive, Discovery, Development, Delivery). We will present numerical and thematic summaries of the knowledge matrix. A qualitative content analysis is out of the scope of this protocol.

**Ethics and dissemination** The scoping review process will only involve identification, selection and analysis of documents available for use in the public domain, and will not include any personal information on individuals, therefore ethical approval is not required. The findings will be disseminated through a peer-reviewed publication and stakeholder meetings.

### STRENGTHS AND LIMITATIONS OF THIS STUDY

⇒ This is a novel literature scoping review that aims to identify knowledge gaps and research questions on antimicrobial resistance (AMR), as the first phase in a stepwise process for developing a priority AMR Global Research Agenda, in consultation with the WHO Steering Group for the AMR Global Research Agenda, and a large group of external experts, policymakers and stakeholders.

⇒ Adopting the modified standardised method for setting priorities in health research developed by the Child Health and Nutrition Research Initiative (CHNRI), the scoping review will populate a predefined conceptual framework ('knowledge matrix') to capture and categorise research questions related to the description of AMR burden and drivers, and the delivery, development and/or discovery of interventions for AMR prevention, diagnosis and/or care and treatment.

⇒ The identification and synthesis of data will include WHO documents, grey literature and systematic reviews published in 2012–2021, to capture the most important research questions and knowledge gaps and inform the further consultation process.

## INTRODUCTION
### Background

The global rise in antimicrobial resistance (AMR) is considered one of the greatest public health threats, with a growing and disproportionate impact in low-income and middle-income countries (LMICs).[1] One of the major drivers of AMR is antimicrobial misuse and overuse both in humans and animals for human consumption.[2] Lack of access to clean water and sanitation in low-resource settings and inadequate infection prevention and control promotes the spread of microbes, including those resistant to antimicrobial treatment. Access to life-saving drugs targeting the emerging resistant pathogens remains an issue in LMICs, while much has been written about the dire state of the

pipeline of new antibacterial drugs. These realities occur in a wider context of lack of access to quality healthcare in many settings. Key strategies to address AMR are to reduce the need for antibiotics (through prevention of infections), to reduce the use of antibiotics (by better use of diagnostics and antibiotic stewardship) and to find new approaches for disease prevention and treatment.

To address the serious public health threat posed by AMR, WHO coordinated the development of a Global Action Plan on AMR in 2015, which in turn has been translated into National Action Plans.[3] To accelerate their implementation, there is a need to expand the evidence-base on AMR burden, impact and interventions, including their prioritisation, cost-effectiveness, financing and knowledge on how to bring existing interventions at scale.[4–6] Since their implementation has been further set back due to the COVID-19 pandemic, now more than ever, there is an urgent need to revitalise work on AMR to control this silent pandemic.[7] Appropriate targeted research is needed to make substantial progress in attaining the 2030 Sustainable Development Goals and to provide the evidence-base in support of successful implementation of interventions to prevent, diagnose and manage drug-resistant infections.

The WHO is developing a Global Research Agenda for AMR that aims to provide a global and transparent assessment of knowledge gaps related to prevention and control of resistant bacterial—including *Mycobacterium tuberculosis*—and fungal infections in the human health sector. This agenda is expected to catalyse investment and focus scientific interest among researchers, donors and public health professionals towards the generation of new and critical evidence to address knowledge gaps for effective implementation of global, regional and national policies on AMR, particularly LMICs, in the timeframe 2023–2030. The evidence generated will inform prioritisation of interventions within AMR National Action Plans. This work focuses on the human health sector only, and will complement the WHO One Health Priority Research Agenda for AMR, being developed in parallel, which focuses on the interface between humans, animals, plants and their shared environment. The first step in the stepwise development of the Agenda is to conduct a scoping literature review to identify existing research questions and knowledge gaps on AMR; this paper outlines the protocol of the scoping review.

### Aim and objectives

To conduct a scoping literature review to identify existing research questions and knowledge gaps on AMR in the human health sector focusing on:
1. Burden of resistant bacterial (including *M. tuberculosis*) and fungal infections and factors associated with and predictive of AMR emergence and transmission;
2. New or improved interventions, technologies and tools that are associated with improved AMR prevention, diagnosis, care and treatment and best ways to deliver these.

## METHODS AND ANALYSIS
### Conceptual framework and scope

The development of the AMR Global Research Agenda will follow the WHO methodological framework *Guidance for undertaking a research priority setting exercise*,[8] which includes a modified version of the standardised method for setting priorities in health research developed by the Child Health and Nutrition Research Initiative (CHNRI).[9 10] The CHNRI framework proposes a flexible, systematic approach to listing a large number of research options and research questions, and allows for a prioritisation exercise that is systematic, transparent, legitimate and fair, scientifically rigorous and replicable. Evidence needed should be clearly framed by the research options or questions, and the effect of its translation and implementation into interventions should be straightforward. The CHNRI framework uses the '4D framework' of 'description', 'delivery', 'development' and 'discovery' research.[9 10] Each identified research question will thus be classified and aggregated to populate a 'knowledge matrix', composed of three focus areas or 'themes' (ie, Prevention, Diagnosis and Care and Treatment) and four cross-cutting domains (Descriptive, Discovery, Development, Delivery), as described in table 1.

To populate the 'knowledge matrix' as a first step in this iterative process, we will conduct a scoping review of existing research questions through reviews of published and grey literature. The scoping review will also be informed by inputs from the WHO Steering Group for the AMR Global Research Agenda (SG) composed of technical experts in areas relevant to AMR, including AMR National Action Plans, Monitoring and Evaluation; surveillance and antimicrobial use; stewardship and awareness; antimicrobial research and development (R&D); WHO essential medicines; diagnostics; infection prevention and control (IPC); water, sanitation, hygiene and health (WASH); vaccine/product and delivery research; the Special Programme for Research and Training In Tropical Diseases (TDR); food safety/foodborne diseases; sexually transmitted infections; newborn health; and tuberculosis (TB) prevention, diagnosis, treatment, care and innovation. The scope of the scoping review is summarised in box 1.

### Protocol design

This protocol follows the methodological framework for scoping reviews developed by Arksey and O'Malley[11] and Levac *et al*[12] through six stages, as well as the guidance developed by the Joanna Briggs Institute to standardise the conduct of scoping reviews.[13] Due to the iterative nature of a scoping review, protocol deviations during the process may be necessary. Any discrepancies will be detailed and justified in the Methods section of the final report. The scoping review is expected to be conducted between November 2021 and April 2022, and includes the following six stages.

**Table 1**  Knowledge matrix

| | | Themes | | |
|---|---|---|---|---|
| | | **Prevention** | **Diagnosis** | **Care and treatment** |
| **Cross-cutting domains ('4Ds')** | **Descriptive** | | | |
| | **Delivery** | | | |
| | **Development** | | | |
| | **Discovery** | | | |

Prevention: prevention of AMR in hospital-acquired and community-acquired infections.
Diagnosis: diagnosis of infections caused by microorganisms resistant to antimicrobials, including both pathogen identification and antibiotic susceptibility (AMR) testing.
Care and treatment: care and treatment of hospital-acquired and community-acquired infections caused by antimicrobial-resistant microorganisms.
Descriptive: improve understanding of AMR epidemiology, burden and factors associated with and predictive of AMR emergence and transmission (epidemiological research).
Delivery: optimise and improve the delivery of existing interventions (implementation research, operations research and/or health policy and systems research).
Development: improve existing interventions by reducing costs, or optimising uptake, impact, sustainability and feasibility.
Discovery: new tools and interventions (new medicines, diagnostics, vaccines or other interventions).
AMR, antimicrobial resistance.

## Stage 1: defining the scope and main research question of the scoping review

The purpose of the scoping review is to map the literature and identify the key research questions and knowledge gaps that inform the research priority setting exercise to develop the AMR Global Research Agenda. This scoping review addresses the following broad question: 'What are the existing research questions and knowledge gaps on the description, delivery, development and/or discovery of interventions for AMR prevention, diagnosis and/or care and treatment in the human health sector, at a global level?'

## Stage 2: searching for relevant documents

We plan to search several information sources of published and grey literature to identify relevant documents as listed in box 2.

All searches will be jointly developed and conducted by an experienced WHO librarian, the authors and content experts in the SG. We will manually search the grey literature in organisations' websites and google, and consider any relevant documents as suggested by the SG. The full search terms for the bibliographic databases are available in online supplemental table 1. For the non-general data sources, we will use less restrictive terms to be able to identify relevant documents. We will search reference lists of included documents for any additional potentially relevant references. The searches will be restricted to the past 10 years (January 2012 to December 2021) and the English language. After the searches, the SG will review the search results to determine if the documents identified represent all relevant resources.

## Stage 3: document selection

This stage will be an iterative process involving searching the literature, refining the search strategy and reviewing articles for eligibility and inclusion. The study team will meet to discuss study inclusion and exclusion criteria at the beginning of the process, and will discuss challenges and uncertainties related to study selection along the review process. Unlike systematic reviews, in scoping reviews, inclusion and exclusion criteria may be adjusted during the search and selection process, once familiarity with the literature has been gained. For grey literature, a first reviewer will read all full-text articles for eligibility, and a second reviewer will separately read at least 20% of all full-text articles; if the agreement is <80%, the second reviewer will dual-read all full-text articles. For bibliographic databases, a first reviewer will screen all titles and abstracts for eligibility, and a second reviewer will separately screen at least 20% of all titles and abstracts; if the agreement is <80%, a second reviewer will dual-screen all titles and abstracts. Then, a first reviewer will read all included full-text articles for eligibility, and a second reviewer will separately read at least 20% of all full-text articles; if the agreement is <80%, a second reviewer will dual-read all full-text articles. We will resolve disagreements on study selection by consensus and discussion with other reviewers as needed. For grey literature, duplicate documents will be removed in Mendeley manually and using the automated tool. For bibliographic databases, duplicate documents will be removed in EndNote using the Bramer method.[14] Documents that are superseded with a more recent version are excluded (and the most recent is included).

Document inclusion criteria:
1. Document describes one or more knowledge gaps or research questions on AMR, including priorities, framework, components, elements or steps for the description (ie, epidemiology, burden and drivers), delivery, development and/or discovery of tools, products or interventions for AMR prevention, diagnosis and/or care and treatment.

## Box 1    Scope of the scoping review

The scoping review
⇒ Is global in its scope;
⇒ Focuses exclusively on AMR in the human health sector;
⇒ Focuses on priority research questions that can and should be addressed in the timeframe 2023–2030 (SDGs);
⇒ Includes relevant publications published in the past 10 years (covering the period leading up to the launch of the Global Action Plan for AMR in 2015, and thereafter);[3]
⇒ Is based on a 'knowledge matrix' composed of three focus areas or 'themes' (Prevention, Diagnosis and Care & Treatment) and four cross-cutting domains (Descriptive, Discovery, Development, Delivery).
⇒ The following disaggregation is considered when categorising the research questions:
⇒ Populations (eg, adults, children, neonates, vulnerable groups);
⇒ Settings (eg, community, primary care, hospitals);
⇒ Geographical/socioeconomic context (eg, low-income and middle-income countries; high-income countries);
⇒ Syndrome (eg, bloodstream infections, etc);
⇒ Antibiotic-microorganism ('drug-bug') combinations.
⇒ Will not rank or prioritise any of the included pathogens for the purpose of this review (see below):
⇒ Focuses on the WHO global priority list of antibiotic-resistant bacteria:[16]
⇒ *Acinetobacter baumannii*: carbapenem-resistant;
⇒ *Pseudomonas aeruginosa*: carbapenem-resistant;
⇒ Enterobacteriaceae (including *Klebsiella pneumonia*, *Escherichia coli*, Enterobacter spp., Serratia spp., Proteus spp. and Providencia spp., Morganella spp.): carbapenem-resistant, third-generation cephalosporin-resistant;
⇒ *Enterococcus faecium*: vancomycin-resistant;
⇒ *Staphylococcus aureus*: methicillin-resistant, vancomycin-intermediate and vancomycin-resistant;
⇒ *Helicobacter pylori*: clarithromycin-resistant;
⇒ Campylobacter: fluoroquinolone-resistant;
⇒ Salmonella spp.: fluoroquinolone-resistant;
⇒ *Neisseria gonorrhoeae*: third-generation cephalosporin-resistant, fluoroquinolone-resistant;
⇒ *Streptococcus pneumoniae*: penicillin-non-susceptible;
⇒ *Haemophilus influenzae*: ampicillin-resistant;
⇒ Shigella spp.: fluoroquinolone-resistant.
⇒ Focuses on drug resistant critical fungi, that is,:
⇒ Aspergillus spp. (especially *A. fumigatus* and *A. flavus*)
⇒ Candida spp. (especially *C. albicans* and *C. auris*).
⇒ Focuses on drug resistant *Mycobacterium tuberculosis*, with the following definitions:
⇒ RR-TB: rifampicin-resistant tuberculosis;
⇒ MDR-TB: resistance to at least isoniazid and rifampicin;
⇒ Pre-XDR-TB: MDR/RR-TB and also resistant to any fluoroquinolone;
⇒ XDR-TB: pre-XDR-TB and also resistant to at least one additional Group A drug (comprises levofloxacin, moxifloxacin, bedaquiline and linezolid).

AMR, antimicrobial resistance; HIC, high-income country; LMICs, low-income and middle-income countries; MDR-TB, multidrug-resistant tuberculosis; Pre-XDR-TB, pre-extensively drug-resistant tuberculosis; RR-TB, rifampicin-resistant tuberculosis; SDGs, sustainable development goals; XDR-TB, extensively drug-resistant tuberculosis.

## Box 2    List of information sources to be searched for the scoping review

⇒ WHO-authored documents and research gaps identified through WHO guidelines processes in areas relevant to AMR
⇒ Target: relevant documents including strategies, global action plans, meeting reports, policy briefs, technical documents and resolutions.
⇒ Approach: we will manually search the WHO Institutional Repository for Data Sharing (IRIS), Data Platform, The Global Health Observatory and WHO Observatory for Health R&D.
⇒ Grey literature relevant to AMR
⇒ Target: relevant position papers, reports, strategies, guidelines and evaluations from national or international agencies and AMR stakeholders.
⇒ Approach: first, we will conduct a global mapping of stakeholders working with AMR policy, advocacy, innovation and research, surveillance, based on the ReAct reference,[17] organisational websites and feedback from the WHO Steering Group for the AMR Global Research Agenda. Second, we will perform a targeted manual search of organisational websites. Third, we will supplement the search above with a google search including the term 'filetype:pdf' and 'antimicrobial resistance' (and synonyms).
⇒ Published systematic reviews on AMR research gaps and priorities
⇒ Target: relevant systematic reviews.
⇒ Approach: systematic search of the relevant bibliographic databases, ie, PubMed, Embase and Web of Science (full search terms in online supplemental table 1).
⇒ Documents suggested by WHO Steering Group for the AMR Global Research Agenda
⇒ Target: relevant scientific papers, editorials, reviews and grey literature.
⇒ Approach: targeted requests to the WHO Steering Group members to share all documents relevant to their specific field of work.

AMR, antimicrobial resistance; R&D, research and development.

1. Document only describes individual research studies or case reports;
2. Document is not related to bacteria included in the WHO priority pathogen list, *M. tuberculosis* or critical fungi (eg, parasites, viruses);
3. Document is related to AMR in the non-human sectors;
4. Document has no identifiable authors, publisher and year of publication.

### Stage 4: extracting the data

Data elements will be extracted from included documents using a data extraction form in Microsoft Excel, which will be jointly developed by the research team and updated during the iterative extraction process ('charting the data'). Two reviewers will independently test the data extraction form on 5–10 documents to ensure all relevant results are extracted and that their approach to extraction is consistent with the research question and purpose. The content of the included documents will be read in-depth to identify research questions and knowledge gaps within the scope of the

2. Document has a global or regional application and/or relevance for LMICs.
   Document exclusion criteria:

scoping review. A single reviewer will extract the data, and translate each research gap, need or question into one or more research questions that are specific for the attributes expressed in the source text. We will use a 'narrative review' or 'descriptive analytical' method to extract information from each document. We will extract explicitly stated research questions, and we will also use relevant text extracts to formulate additional research questions. This includes results of previous research priorities exercises if available. Each research question will be annotated with relevant attributes (online supplemental table 2). The extracted research questions and annotations will be reviewed and refined by at least two other team members. The research team will meet weekly to review the research questions identified and ensure data extraction is consistent with the research question and purpose. The whole exercise will result in a database of annotated research questions. Document authors will not be contacted.

## Stage 5: collating, summarising and reporting results

To provide an overview of the breadth of the literature, we will present numerical and thematic summaries of the key characteristics and categories of included documents and research questions, using the thematic construction of the 'knowledge framework' (box 2). The results may be presented as a 'map' of the data in a logical, diagrammatic or tabular form, and/or in a descriptive format that aligns to the overall study purpose and scope of the review. We will seek to minimise redundancies by deleting and/or merging research questions into higher-level research avenues where appropriate and feasible. The plan for presenting the results will be further refined toward the end of the review when the researchers have gained the greatest awareness of the contents of the included studies. However, an in-depth qualitative content analysis or synthesis, and discussion of implications for future research, practice and policy are outside the scope of this scoping review. Furthermore, we will not perform a formal assessment of the methodological quality of the included documents as the purpose of this scoping review is to achieve a broad and in-depth description of existing research questions and knowledge gaps around AMR.

We will report in compliance with the Preferred Reporting Items for Systematic Reviews and Meta-Analyses extension for Scoping Reviews including a flow diagram for the scoping review process and a completed checklist.[15]

## Stage 6: consulting with stakeholders

The scoping review process will be conducted in regular consultation with the SG to inform and validate the results. Later in the agenda-setting process, the findings of the scoping review will form a basis for various rounds of expert and stakeholder consultations for ranking, prioritising and validation towards further finalisation of the WHO Global Research Agenda for AMR.

### Patient and public involvement statement

The patient and public were not involved in the development of this study protocol.

## ETHICS AND DISSEMINATION

The scoping review process will only involve identification, selection and analysis of documents available for use in the public domain, and will not include any personal information on individuals, therefore ethical approval is not required. The scoping review will be done in consultation with the SG and disseminated through a peer-reviewed publication. This scoping review is the first step in a stepwise process for developing a priority AMR Global Research Agenda, which will be done in consultation with global, regional and national stakeholders who are the end-users of the generated knowledge and disseminated through a final report.

**Acknowledgements** The authors would like to acknowledge the contributions and suggestions of the WHO Steering Group for the AMR Global Research Agenda in the Human Health Sector: Kitty Van Weezenbeek, Sarah Paulin, Barbara Tornimbene, Tine Jorgensen, Peter Beyer, Thomas Joseph, Benedetta Allegranzi, Nebiat Gebreselassie, Mateusz Hasso-Agopsowicz, Rob Terry, Benedikt Huttner, Bruce Gordon, Francis Moussy, Luz De Regil, Sachiyo Yoshida, Teodora Wi. The authors also acknowledge the contributions and suggestions of other WHO colleagues: Anand Balachandran, Verica Ivanovska, Kate Medlicott, Arno Muller, Maarten van der Heijden, Carmem Pessoa and Matteo Zignol, and the support of the WHO librarians Tomas Allen and Kavita Kothari.

**Contributors** SB and AC conceptualised the project. RLH, AC and SB developed the protocol and search strategies. RLH drafted the manuscript, with critical input from AC, KSA and SB. KSA designed and tested the data extraction tools. All authors critically reviewed the paper for important intellectual content and approved the final version of the manuscript.

**Funding** This work is part of the project entitled 'Global Research Agenda for Antimicrobial Resistance in the Human Health Sector' initiated and funded by WHO (no award/grant number).

**Competing interests** SB and AC are WHO employees. RLH is a consultant for this work and is also supported by the Wellcome Trust (106680/Z/14/Z). The views expressed are those of the authors and not necessarily those of WHO, Wellcome Trust, or any of the institutions mentioned.

**Patient and public involvement** Patients and/or the public were not involved in the design, or conduct, or reporting or dissemination plans of this research.

**Patient consent for publication** Not applicable.

**Provenance and peer review** Not commissioned; externally peer reviewed.

**ORCID iD**
Raph L Hamers http://orcid.org/0000-0002-5007-7896

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
