## [Reviewer comments · BMJ Open]

ARTICLE DETAILS

TITLE (PROVISIONAL)	Developing a priority global research agenda for antimicrobial resistance in the human health sector: protocol for a scoping review
AUTHORS	Hamers, Raph Leonardus; Cassini, Alessandro; Asadinia, Koe Stella; Bertagnolio, S

VERSION 1 – REVIEW

REVIEWER	Rzewuska, Magdalena University of Aberdeen Institute of Applied Health Sciences, Health Services Research Unit
REVIEW RETURNED	27-Jan-2022

GENERAL COMMENTS	It's an excellent and much needed piece of research, using a novel and robust method, transparently and comprehensively outlined in this protocol, and to be reported using relevant reporting guidance. Below are my few suggestions to improve clarity of reporting, for the authors' consideration: Main comments The authors stated that "the protocol follows the methodological framework for scoping reviews developed by" Arksey and O'Malley and Levac et al. through six stages, as well as the guidance developed by the Joanna Briggs Institute to standardize the conduct of scoping reviews.[12]" Reference [12] is an 88-pages document, which doesn't seem to mention Arksey and O'Malley's. I think that for the sake of reproducibility<="" span="" style="font-family:Arial">, the readers will appreciate a supplement material beefily outlining that methodological framework, including how the six stages of Arksey method and the guidance were jointly used. The authors stated that "the purpose of the scoping review is to map the literature and identify the key research questions and knowledge gaps" (line 29-31). Having read supplementary Table 2, it's unclear to me what is an outcome(s) in this study and what types of data will be considered relevant to that outcome(s) (e.g. results of research priority exercises, any explicit forms of identification of research priorities and/or knowledge gaps, research implications stated anywhere in the included article/document). At the core of this issue is a question of whether this work will involve>identifying only reported knowledge gaps and research questions (as a primary/secondary outcome and/or stated research implication) or the authors will draw inferences from reviewed evidence about knowledge gaps and research questions. I suspect it's the latter, but it should be clarified.
--

	In terms of the scope, I think deciding what constitutes an AMS intervention (currently undecided) will likely pose a major challenge. There are direct “micro” interventions, such as hand washing, and/or “macro” policy interventions, such as legislation, global control of drug availability, tax/subsidy. The authors mentioned key strategies to address AMR such as “reduc[ing] the need for antibiotics (through prevention of infections), to reduce the use of antibiotics (by better use of diagnostics and antibiotic stewardship), and to find new methods of disease prevention and treatment”. This sounds like they may be interested also in a wide range of studies on delivery and development of indirect interventions, including antibiotic stewardship studies, which often don’t report effects on AMR and may not even focus on any specific pathogen or pathogens. To account for this likely major challenge, I wonder if it may help to clarify this in the protocol or to state that unlike systematic reviews, in scoping reviews, inclusion and exclusion criteria are developed post hoc, once familiarity with the literature has been gained (Arksey & O’Malley, 2005), which is quite different from protocol deviation. Minor comments The rationale for this work is generating new evidence, particularly for LMIC, which are disproportionately more affected by the AMR problem than HIC, and yet the authors plan to include evidence from both LMIC and HIC. Please state the reason for this decision. Please state the reason for reviewing evidence from only the past ten years (even if the reason is as simple as this time scale being previously or commonly used for similar work). BMJ open protocol papers should report planned or ongoing studies. The dates of the study should be included in the manuscript.
--	---

REVIEWER	Arega, B Yekatit 12 Hospital Medical College , internal medicine
REVIEW RETURNED	08-Feb-2022

GENERAL COMMENTS	This a very interesting review on the top global agenda. I have only minor comments marked on the manuscript.
---

VERSION 1 – AUTHOR RESPONSE

Reviewer 1

Dr. Magdalena Rzewuska, University of Aberdeen Institute of Applied Health Sciences

Comments to the Author:

It’s an excellent and much needed piece of research, using a novel and robust method, transparently and comprehensively outlined in this protocol, and to be reported using relevant reporting guidance.

Author response: We thank the Reviewer for their positive feedback.

Below are my few suggestions to improve clarity of reporting, for the authors’ consideration:

Main comments

1. The authors stated that “the protocol follows the methodological framework for scoping

reviews developed by” Arksey and O’Malley and Levac et al. through six stages, as well as the guidance developed by the Joanna Briggs Institute to standardize the conduct of scoping reviews.[12]” Reference [12] is an 88-pages document, which doesn’t seem to mention Arksey and O’Malley’s. I think that for the sake of reproducibility, the readers will appreciate a supplement material beefily outlining that methodological framework, including how the six stages of Arksey method and the guidance were jointly used.

Author response:

In 2005, Hilary Arksey & Lisa O’Malley provided the foundation of a methodological framework for scoping studies, which distinguished and identified five subsequent stages plus an optional consultation stage (Hilary Arksey & Lisa O’Malley (2005) Scoping studies: towards a methodological framework, *International Journal of Social Research Methodology*, 8:1, 19-32, DOI: 10.1080/1364557032000119616). In 2010, Levac et al. provided specific recommendations to clarify and enhance the methodology for each stage of the Arksey and O’Malley framework (Levac et al.: Scoping studies: advancing the methodology. *Implementation Science* 2010 5:69. doi:10.1186/1748-5908-5-69.) Our scoping review follows the six-stage approach as described in these guidance documents, and for clarification we have now included those two references. We opted not to include the supplement material to avoid redundancy with the main text.

Peters et al. (Reference 12 in the original submission) is a 6-page paper summarizing the guidance for the standardized conduct of systematic scoping reviews developed by members of the Joanna Briggs Institute Collaborating Centres (Peters MDJ, Godfrey CM, Khalil H, et al. Guidance for conducting systematic scoping reviews. *JBI Evid Implement* 2015;13.https://journals.lww.com/ijebh/Fulltext/2015/09000/Guidance_for_conducting_systematic_scoping_reviews.5.aspx). We are unsure which “88-pages document” the reviewer is referring to.

1. The authors stated that “the purpose of the scoping review is to map the literature and identify the key research questions and knowledge gaps” (line 29-31). Having read supplementary Table 2, it’s unclear to me what is an outcome(s) in this study and what types of data will be considered relevant to that outcome(s) (e.g. results of research priority exercises, any explicit forms of identification of research priorities and/or knowledge gaps, research implications stated anywhere in the included article/document). At the core of this issue is a question of whether this work will involve identifying only reported knowledge gaps and research questions (as a primary/secondary outcome and/or stated research implication) or the authors will draw inferences from reviewed evidence about knowledge gaps and research questions. I suspect it’s the latter, but it should be clarified.

Author response:

We aim to comprehensively identify and aggregate all relevant existing research questions, research priorities and knowledge gaps from the grey and bibliographic literature, spanning the full scope as described in the protocol. This includes results of previous research priorities exercises if available. In doing so, we will extract explicitly stated research questions, and the authors will also formulate research questions based on relevant text extracts in the included documents. The whole exercise will result in a database of research questions, annotated by extracted information according to the protocol. Annotations will allow analysis and stratification of research questions. During the consolidation stage, we will seek to minimize redundancies by deleting and/or merging the pertinent research questions into higher-level questions where appropriate. We clarified this in the manuscript.

1. In terms of the scope, I think deciding what constitutes an AMS intervention (currently undecided) will likely pose a major challenge. There are direct “micro” interventions, such as hand washing, and/or “macro” policy interventions, such as legislation, global control of drug

availability, tax/subsidy. The authors mentioned key strategies to address AMR such as “reduc[ing] the need for antibiotics (through prevention of infections), to reduce the use of antibiotics (by better use of diagnostics and antibiotic stewardship), and to find new methods of disease prevention and treatment”. This sounds like they may be interested also in a wide range of studies on delivery and development of indirect interventions, including antibiotic stewardship studies, which often don’t report effects on AMR and may not even focus on any specific pathogen or pathogens. To account for this likely major challenge, I wonder if it may help to clarify this in the protocol or to state that unlike systematic reviews, in scoping reviews, inclusion and exclusion criteria are developed post hoc, once familiarity with the literature has been gained (Arksey & O’Malley, 2005), which is quite different from protocol deviation.

Author response:

We appreciate the Reviewer’s thoughtful comments and are in full agreement that this scoping review is an ambitious undertaking, in which we will encounter a broad range of studies, methodologies, interventions, including research questions with different levels of detail, scope and focus. As suggested by the Reviewer, we have included a clarifying sentence on the possibility for post-hoc adjustments to the eligibility criteria and other aspects of the methodology.

Minor comments

1. The rationale for this work is generating new evidence, particularly for LMIC, which are disproportionately more affected by the AMR problem than HIC, and yet the authors plan to include evidence from both LMIC and HIC. Please state the reason for this decision.

Author response: In accordance with the WHO’s mandate, the global research agenda for AMR will have a global scope with relevance both for HIC and LMIC. If there are important lessons to be learned from the rich HIC literature, these will be useful to augment the scoping review that is relevant globally as well as for LMIC. The research questions will also be disaggregated by HIC vs LMIC in the analysis stage.

1. Please state the reason for reviewing evidence from only the past ten years (even if the reason is as simple as this time scale being previously or commonly used for similar work).

Author response: We limited the review of evidence to the most recent 10 years to be aligned with the launch and implementation of the WHO Global Action Plan for AMR in 2015 (permitting a reasonable lead-in period of approximately two years). The GAP has been a great accelerator of the research agenda and has been followed by work that is most relevant for this scoping review. Older literature may no longer be relevant or may be superseded by newer versions. We included clarifying statements.

1. BMJ open protocol papers should report planned or ongoing studies. The dates of the study should be included in the manuscript.

Author response: The scoping review is expected to be conducted between December 2021 and April 2022. We included the timeline and dates of the study in the Methods.

Reviewer 2:

Dr. B Arega, Yekatit 12 Hospital Medical College

This a very interesting review on the top global agenda. I have only minor comments marked on the manuscript.

Author response: We thank the Reviewer for their positive feedback.

1. Abstract>Methods and analysis: This part should good to be shorten

Author response:

We have attempted to shorten this section.

1. Introduction: Background (page 5) Line 17-33: need reference, It looks the idea taken from other literatures

Author response: This scoping review is the first step of the stepwise development of a WHO Global Research Agenda for AMR, and there is no reference yet for this newly started initiative. This has been clarified in the text.

1. Background (page 6) Line 10-22, it has to be referenced

Author response: The methodology, specifically CHNRI framework, described in that section has been referenced. Unfortunately, we are unclear on what more the Reviewer would like to see referenced.

1. Table 1 footnotes: "Diagnosis": the definition is not clear. Is to mean taking the risk factors for AMR, then perform pathogen isolation and AMR ?. Could it be effective way to control AMR ?

Author response: We clarified the definition in Table 1 footnotes.

1. METHODS AND ANALYSIS: Conceptual framework and scope (Page 9): Please define What id D and R. considers others similar too.

Author response: We explained all acronyms throughout the manuscript.

1. Panel (page 9): "Will not consider prioritization of specific pathogens": hard to understand the difference to this sentence "focused with the WHO global list of AMR pathogens"

Author response: This means that the scoping review will consider all the target pathogens equally, we will not rank or prioritize them as part of this scoping exercise. We clarified this in the panel.

VERSION 2 – REVIEW

REVIEWER	Rzewuska, Magdalena University of Aberdeen Institute of Applied Health Sciences, Health Services Research Unit
REVIEW RETURNED	04-Apr-2022

GENERAL COMMENTS	The authors have carefully considered my comments. I'm happy with the responses provided. I have no further comments to offer. Good luck with this study.
---

REVIEWER	Arega, B Yekatit 12 Hospital Medical College , internal medicine
REVIEW RETURNED	10-Apr-2022

GENERAL COMMENTS	The concerns I raised in the first draft are well addressed
---